Thoracic SMARCA4-deficient tumors: a clinicopathological analysis of 52 cases with SMARCA4-deficient non-small cell lung cancer and 20 cases with thoracic SMARCA4-deficient undifferentiated tumor

Zhou Ping
Fu Yiyun
Tang Yuan
Jiang Lili
Wang Weiya 151422303@qq.com
Department of Pathology, West China Hospital, Sichuan University , Chengdu, Sichuan , China
Zhang Xin
Electronic publication date: 2024 Feb 16
Publication date: 2024
Volume: 12
Electronic Location ID: e16923
Received 2023 Oct 8; Accepted 2024 Jan 19
Copyright: © 2024 Zhou et al.
Copyright year: 2024
Copyright holder: Zhou et al.
License: This is an open access article distributed under the terms of the Creative Commons Attribution License, which permits unrestricted use, distribution, reproduction and adaptation in any medium and for any purpose provided that it is properly attributed. For attribution, the original author(s), title, publication source (PeerJ) and either DOI or URL of the article must be cited.
License URL: https://creativecommons.org/licenses/by/4.0/

Keywords: SWI/SNF, SMARCA4, BRG1, Undifferentiated tumor, NSCLC

Funding: Sichuan University No. 2019HXFH034 and ZYJC21074 Sichuan Science and Technology Program No. 2021YJ0117 The writing of this article was supported by 1.3.5 Project for Disciplines of Excellence-Clinical Research Incubation Project, West China Hospital, Sichuan University (No. 2019HXFH034 and ZYJC21074) and Sichuan Science and Technology Program (No. 2021YJ0117). The funders had no role in study design, data collection and analysis, decision to publish, or preparation of the manuscript.

==============================
Background

Thoracic SMARCA4-deficient undifferentiated tumor (SMARCA4-UT) is a distinct clinicopathological entity with an aggressive clinical course. Additionally, SMARCA4/BRG1 deficiency can be observed in a few patients with non-small cell lung cancer (NSCLC). We aimed to compare the clinicopathological, immunohistochemical and prognostic features of SMARCA4-deficient NSCLC (SMARCA4-dNSCLC) with those of thoracic SMARCA4-UT.

Methods

Patients with BRG1-deficient tumors in the lung or thorax were enrolled in the study from the Department of Pathology of West China Hospital, Sichuan University, from January 2014 to June 2022. We retrospectively collected the clinicopathological and immunohistochemical features and outcomes of these patients.

Results

Seventy-two patients had tumors in the lung or thorax with BRG1-deficient expression, including 52 patients with SMARCA4-dNSCLC and 20 patients with thoracic SMARCA4-UT. Among the patients with SMARCA4-dNSCLC, 98.1% were male, 85.7% were smokers, and 79.5% (35/44) had tumor-node-metas­tasis (TNM) III-IV tumors. Among the patients with thoracic SMARCA4-UT, all were males who smoked, and 93.75% (15/16) had TNM III-IV tumors. Pure solid architecture and necrosis were the predominant pathological features. Rhabdoid morphology was observed in some SMARCA4-dNSCLCs (10/52, 19.2%) and thoracic SMARCA4-UTs (11/20, 55%). In most patients with thoracic SMARCA4-UT, the tumors exhibited scattered weak expression or negative expression of epithelial markers, and positive expression of CD34 and Syn. Overall survival (OS) and progression-free survival (PFS) were not significantly different between patients with SMARCA4-dNSCLC and patients with thoracic SMARCA4-UT (p = 0.63 and p = 0.20, respectively).

Conclusions

Thoracic SMARCA4-DTs include SMARCA4-dNSCLC and thoracic SMARCA4-UT. Both have overlapping clinicopathological features and poor prognosis. We hypothesize that thoracic SMARCA4-UT may be the undifferentiated or dedifferentiated form of SMARCA4-dNSCLC. However, further studies with larger cohorts and longer follow-up periods are needed.

Introduction

Mammalian switch/sucrose nonfermentable (SWI/SNF) complexes function as epigenetic regulators and play critical roles in the development of tumors. The impacts of these complexes are extensive; these complexes influence the regulation of cell cycle progression, oncogenic pathways, and metabolism (Masliah-Planchon et al., 2015). Notably, mutations in SWI/SNF subunits are prevalent and occur in 19.6% of all human tumors (Kadoch et al., 2013). SMARCA4 is a catalytic ATPase subunit of the SWI/SNF complex that encodes Brahma-related gene 1 (BRG1). This specific subunit has been shown to be a tumor suppressor (Oike et al., 2013; Marquez-Vilendrer et al., 2016). The complicated functions of SWI/SNF complexes, coupled with the prevalence of subunit mutations, highlight their importance in the complicated landscape of tumorigenesis. Specifically, the role of BRG1 as a tumor suppressor further emphasizes the molecular mechanisms that are involved in cancer development and regulation. SMARCA4 mutations are found in a variety of malignant tumors, including small cell carcinoma of the ovary, hypercalcemic type (SCCOHT), lung adenocarcinoma, undifferentiated carcinoma of the gastrointestinal tract, medulloblastoma, Burkitt lymphoma, prostate cancer, breast cancer and uterine sarcoma (Ramos et al., 2014; Imielinski et al., 2012; Robinson et al., 2012; Love et al., 2012; Kolin et al., 2020; Bai et al., 2013; Muthuswami et al., 2019; Li et al., 2021; Bell et al., 2016; Chang et al., 2022).

The 5th edition of the 2021 World Health Organization (WHO) classification of thoracic tumors defines thoracic SMARCA4-deficient undifferentiated tumor (SMARCA4-UT) as a new entity of rare malignant tumor with an aggressive clinical course (Nicholson et al., 2021). Thoracic SMARCA4-UTs are high-grade malignant neoplasms that significantly affect the thorax of adults, are characterized by an undifferentiated or rhabdoid phenotype, and displays a deficiency of SMARCA4 (BRG1). SMARCA4 deficiency can be observed in a subset of non-small cell lung cancers (NSCLCs) (Agaimy et al., 2017). Dagogo-Jack et al. (2020) observed SMARCA4 genomic alterations in 8% (117 of 1,422) and 12% (3,188 of 27,281) of NSCLCs in the institutional and Foundation Medicine datasets, respectively. Among 3416 NSCLC patients, approximately 25% had at least one SWI/SNF complex gene mutation, and of these patients, 9% harbored SMARCA4 mutations (Zhu et al., 2021). SMARCA4-deficient NSCLC (SMARCA4-dNSCLC) is a rare subtype of NSCLC that has unique clinicopathological features and a worse prognosis than SMARCA4-intact NSCLC (Bell et al., 2016; Liang et al., 2023).

Immunohistochemical staining for BRG1 is useful for identifying thoracic SMARCA4-deficient tumors (SMARCA4-DTs). Thoracic SMARCA4-DTs include SMARCA4-dNSCLC and thoracic SMARCA4-UT (Decroix et al., 2020). Thoracic SMARCA4-UT tends to be larger and have a worse prognosis than SMARCA4-dNSCLC (Rekhtman et al., 2020). SMARCA4-dNSCLC can be distinguished from thoracic SMARCA4-UT according to their gland architecture, cellular cohesion and positive expression of epithelial markers, including pan-cytokeratin (PCK) and epithelial membrane antigen (EMA). Thoracic SMARCA4-UT exhibits scattered weak or negative expression of epithelial markers (Nicholson et al., 2021), whereas SMARCA4-dNSCLC exhibits strong and/or diffuse expression of these markers. However, limited comparisons of these two distinct clinicopathological entities have been performed. Herein, we retrospectively compared the clinical, pathological, immunophenotypical and prognostic features of thoracic SMARCA4-DTs, and we provide insights into the clinicopathological features of SMARCA4-dNSCLC and thoracic SMARCA4-UT.

Patients and methods

Patients and tumor specimens

Thoracic SMARCA4-DTs were initially diagnosed in routine clinical practice between January 2020 and June 2022. IHC staining for BRG1 was retrospectively performed in poorly differentiated or undifferentiated tumors in lung and thorax from January 2014 to December 2019. Patients with BRG1-deficient tumors of the lung or thorax were enrolled in the study from the Department of Pathology of West China Hospital, Sichuan University, between January 2014 and June 2022. Patients with BRG1-retained tumors of the lung or thorax and patients without BRG1 immunohistochemical staining results were excluded. We retrospectively collected the clinicopathological features, immunohistochemical features and outcomes of these patients. TNM stage was determined according to the 8th lung cancer tumor-node-metastasis (TNM) classification (Lim et al., 2018), and differentiation and the histological subtype were performed according to the 5th edition of the 2021 WHO classification of thoracic tumors (Nicholson et al., 2021). Overall survival (OS) was defined as the time from primary diagnosis to death, and progression-free survival (PFS) was defined as the time from primary diagnosis to tumor recurrence or progression. Ethics approval was obtained from the respective ethics committees of West China Hospital, Sichuan University, China (NO.2022317).

Immunohistochemical staining

All of the specimens were fixed with 10% neutral buffered formalin solution and embedded in paraffin within 24–48 h after surgery and biopsy. Immunohistochemistry was performed on 4-μm whole sections that were generated from formalin-fixed paraffin-embedded tissue blocks. Envision two-step immunohistochemical staining was performed according to previously described methods (Kammerer et al., 2001). Antigen retrieval was performed at 95°C for 16 min in sodium citrate buffer (pH 6.0). An anti-SMARCA4 antibody (anti-BRG1 antibody, 1:200 dilution; clone EPNCIR111A; Abcam, Cambridge, MA, USA) was used to stain all the available specimens. Antibodies against epithelial markers, including pan-cytokeratin (PCK) (clone AE1/AE3, 1:200 dilution; ZSGB-BIO, Beijing, China) and epithelial membrane antigen (EMA) (clone GP1.4, 1:150 dilution; ZSGB-BIO, Beijing, China); stem cell markers, including CD34 (clone EP88, 1:200 dilution; ZSGB-BIO, Beijing, China) and spalt-like transcription factor 4 (SALL4) (clone 6E3, ready to use assay; ZSGB-BIO, Beijing, China); lung adenocarcinoma markers, including CK7 (clone EP16, 1:200 dilution; ZSGB-BIO, Beijing, China), TTF-1 (clone 8G7G3/1, 1:150 dilution; ZSGB-BIO, Beijing, China) and Napsin A (clone MX015, 1:200 dilution, MXB); squamous cell carcinoma makers, including p63 (clone UMAB4/4A4, 1:300 dilution; ZSGB-BIO, Beijing, China), p40 (clone ZR8, 1:100 dilution, MXB) and CK5&6 (clone MX040, 1:200 dilution, MXB); neuroendocrine markers, including synaptophysin (Syn) (clone EP158, 1:200 dilution; ZSGB-BIO, Beijing, China) and chromogranin A (CgA) (clone LK2H10, 1:200 dilution, ZSGB-BIO, Beijing, China); the proliferation marker Ki-67 (clone MIB-1, 1:300 dilution; ZSGB-BIO, Beijing, China); SMARCB1 (INI1) (clone 25, ready to use assay; ZSGB-BIO, Beijing, China); nuclear protein in testis (NUT), which is a marker of NUT-middle carcinoma (clone B1, ready to use assay; ZSGB-BIO, Beijing, China); targetable oncogenic therapy proteins, including anaplastic lymphoma kinase-Ventana (ALK-V) (clone D5F3, ready to use assay; Roche, Basel, Switzerland) and c-ros oncogene 1 (ROS1) (clone OTI1A1, ready to use assay; ZSGB-BIO, Beijing, China) were used to stain all the available specimens. The staining results were determined by two pathologists.

Statistical analysis

Statistical analyses were performed by using SPSS for Windows, version 20.0 (SPSS, Inc., Chicago, IL, USA) and GraphPad Prism version 8 (GraphPad Software, La Jolla, CA, USA). The Kaplan‒Meier method was used to analyze OS and PFS, and the log-rank test was performed based on the differences. The log-rank test was used to determine differences in OS and PFS between patients with SMARCA4-dNSCLC and thoracic SMARCA4-UT. The chi-square test was used to analyze the associations of the clinicopathological characteristics of SMARCA4-dNSCLC and thoracic SMARCA4-UT. A p value of <0.05 was considered to indicate statistical significance.

Results

Clinical features

From January 2014 to June 2022, 759 tumor specimens of the lung or thorax with BRG1 immunoexpression were retrospectively rescreened. Ultimately, 72 (9.5%) of the patients who were included in this study had BRG1-deficient expression, according to immunostaining experiments. Of the 72 patients with SMARCA4-DTs, 52 patients were poorly differentiated SMARCA4-dNSCLC, and 20 patients were thoracic SMARCA4-UT. This series of patients included some consultative cases in our institution, and the clinical features and follow-up information for some consultative cases were not available. The clinical characteristics of the included patients are shown in Table 1.

Table 1 Clinical comparisons between SMARCA4-dNSCLC and thoracic SMARCA4-UT.

Characteristics	SMARCA4-dNSCLC
(n = 52)	Thoracic SMARCA4-UT
(n = 20)	P value	
Sex (n, (%))		0.532	
Male	51(98.1)	20 (100)		
Female	1 (1.9)	0		
Median age (range) (y)	59.5 (34–78)	61.0 (39–72)		
Smoking (n, (%))		0.109	
Yes	42/49 (85.7)	16/16 (100)		
No	7/49 (14.3)	0		
Tumor sites (n, (%))			0.053	
Left upper lobe	15 (28.8)	4 (20)		
Right upper lobe	13 (25.0)	2 (10)		
Right lower lobe	7 (13.5)	2 (10)		
Right middle lobe	6 (11.5)	0		
Left lower lobe	3 (5.8)	2 (10)		
Right lung	2 (3.8)	0		
Mediastinum	2 (3.8)	6 (30)		
Pleura	1 (1.9)	2 (10)		
Left lung	1 (1.9)	1(5)		
Lung and chest	1 (1.9)	0		
Chest wall	1 (1.9)	1 (5)		
Surgical treatment (n, (%))		0.165	
Yes	25 (48.1)	6 (30)		
No	27 (51.9)	14 (70)		
Differentiation (n, (%))			
Poor	52 (100)	/		
TNM stage (n, (%))		0.029	
I	5/44 (11.4)	0		
II	4/44 (9.1)	1/16 (6.25)		
III	18/44 (40.9)	2/16 (12.5)		
IV	17/44 (38.6)	13/16 (81.25)		
Note:

SMARCA4-dNSCLC, SMARCA4-deficient non-small cell lung cancer; SMARCA4-UT, SMARCA4-deficient undifferentiated tumor; y, years; TNM, tumor-node-metastasis.

Among the 52 patients with SMARCA4-dNSCLC, 98.1% (51/52) were male, and 85.7% (42/49) were either current or former smokers. The median age at diagnosis was 59.5 years (ranging from 34 to 78 years). A total of 48.1% (25/52) of these patients were treated surgically. The primary tumor sites were mostly located in the upper lobe, with 15 (28.8%) primary tumor sites in the left upper lobe and 13 (25.0%) primary tumor sites in the right upper lobe. In addition, seven (13.5%) were in the right lower lobe, six (11.5%) were in the right middle lobe, three (5.8%) were in the left lower lobe, two (3.8%) were in the right lung, two (3.8%) involved in the mediastinum, one (1.9%) involved in the pleura, one (1.9%) was in the left lung, one (1.9%) involved the lung and chest, and one (1.9%) involved in the chest wall. Pathology revealed the following subtypes: 31 (59.6%) patients were adenocarcinoma, eight (15.4%) patients were large cell carcinoma, one (1.9%) patient was combined large cell neuroendocrine carcinoma (LCNEC) with adenocarcinoma, and 12 (23.1%) patients were NSCLC-not otherwise specified (NSCLC-NOS) due to biopsy. Information about the TNM stage was available for 44 SMARCA4-dNSCLCs, including stage I (5, 11.4%), stage II (4, 9.1%), stage III (18, 40.9%), and stage IV (17, 38.6%).

All 20 of the patients with thoracic SMARCA4-UT were male, and 100% (16/16) of the patients had a history of smoking. The median age at diagnosis was 61.0 years (ranging from 39 to 72 years). Among these patients, 30% (6/20) were treated surgically. The primary tumors were located in the following sites: six (30%) primary tumor sites were in the mediastinum, four (20%) were in the left upper lobe, two (10%) were in the pleura, two (10%) were in the right upper lobe, two (10%) were in the right lower lobe, two (10%) were in the left lower lobe, one (5%) was in the left lung and one (5%) was in the chest wall. Information about the TNM stage was available for 16 thoracic SMARCA4-UTs, including stage II (1, 6.25%), stage III (2, 12.5%), and stage IV (13, 81.25%).

Patients with both SMARCA4-dNSCLC and thoracic SMARCA4-UT had a worse survival. Among the patients with SMACAR4-dNSCLC, 74.5% (35/47) were cancer-related deaths, with a median OS of 7.8 months. Among the patients with SMACAR4-UT, 80% (16/20) were cancer-related deaths, with a median OS of 5.6 months. Kaplan‒Meier analysis showed that OS (p = 0.63, Fig. 1A) and PFS (p = 0.20, Fig. 1B) did not significantly differ between patients with SMARCA4-dNSCLC and thoracic SMARCA4-UT.

Figure 1 Kaplan‒Meier analysis of patients with SMARCA4-dNSCLC and thoracic SMARCA4-UT.

Survival was not significantly different between patients with SMARCA4-dNSCLC and patients with thoracic SMARCA4-UT, as determined by the log-rank test. (A) Overall survival (OS), p value = 0.63; (B) pro­gression-free survival (PFS), p value = 0.20.

Pathological findings

There were 31 resected tumor tissues and 41 biopsies. The pathological findings are summarized in Table 2 and Fig. 2. Histologically, a solid architecture (Figs. 2A and 2E), irregular sheet or nest arrangement, typical infiltration of the surrounding tissues, typical tumor necrosis (Figs. 2B and 2F), and rhabdoid morphology (Figs. 2C and 2G) were the major pathological characteristics of thoracic SMARCA4-DTs. Focal glandular formation (Fig. 2I), desmoplastic stroma (Fig. 2J), focal stromal myxoid changes (Fig. 2K), pale cytoplasm (Fig. 2L) and neutrophil infiltration (Fig. 2M) were the rare pathological characteristics of thoracic SMARCA4-DTs.

Table 2 Pathological comparisons between SMARCA4-dNSCLC and thoracic SMARCA4-UT.

Characteristics	SMARCA4-dNSCLC
(n = 52)	Thoracic SMARCA4-UT
(n = 20)	p value	
Architecture (n, (%))			0.284	
Solid architecture	46 (88.5)	20 (100)		
Solid architecture and glands	4 (7.7)	0		
Scattered tumor cells	2 (3.8)	0		
Necrosis (n, (%))	24 (46.2)	8 (40)	0.638	
Rhabdoid morphology (n, (%))	10 (19.2)	11 (55)	0.003*	
Desmoplastic stroma (n, (%))	4 (7.7)	0	0.202	
Myxoid stroma (n, (%))	4 (7.7)	1 (5)	0.687	
Pale cytoplasm (n, (%))	2 (3.8)	0	0.374	
Neutrophil infiltration (n, (%))	10 (19.2)	3 (15)	0.676	
Notes:

SMARCA4-dNSCLC, SMARCA4-deficient non-small cell lung cancer; SMARCA4-UT, SMARCA4-deficient undifferentiated tumor.

* Statistically significant.

Figure 2 Pathological features of thoracic SMARCA4-deficient tumors.

Typical solid architecture with large nests and sheets of neoplastic cells, with typical tumor necrosis (A and E) (magnification 40×). Tumor cells were cohesive in SMARCA4-dNSCLCs (B), whereas, tumor cells had an epithelioid morphology and were poorly cohesive or noncohesive in thoracic SMARCA4-UTs (F) (magnification 200×). Rhabdoid morphological cells showing abundant eosinophilic cytoplasm and large vesicular nuclei with prominent nucleoli (C and G) (magnification 200×). BRG1 was deficient in SMARCA4-dNSCLCs (D, red arrow) and thoracic SMARCA4-UTs (H, red arrow) as shown by immunostaining, and the inflammatory cells that were used as internal positive controls were positive for BRG1 expression (D and H, blue arrows). (magnification 100×) (I), Focal glandular formation in a few SMARCA4-dNSCLCs. (magnification 100×) (J), Desmoplastic, round tumor-like pattern showing sheets and nests of tumor cells surrounded by a prominent desmoplastic stroma with a variably prominent fibroblastic component. (magnification 100×) (K), Myxoid stromal changes were focal in rare tumors. Tumor cells were arranged in a cord-like manner or floated individually in mucinous pools in these areas. (magnification 200×) (L), Rare tumor cells with abundant pale cytoplasm. (magnification 200×) (M), Obvious neutrophil infiltration among tumor cells. (magnification 200×).

Of the 52 SMARCA4-dNSCLCs, 46 (88.5%) presented a pure solid architecture, four (7.7%) presented with solid architecture and focal glandular formations, and two (3.8%) had few scattered tumor cells. Necrosis was ubiquitously observed in 24 (46.2%) samples, presenting with geographic distribution. One (1.9%) patient was diagnosed with LCNEC combined with adenocarcinoma, which had a solid architecture of neuroendocrine morphology and geographic necrosis. Of the samples, 19.2% (10/52) presented with focal rhabdoid morphology (Fig. 2C). In a few samples (7.7%, 4/52), tumor cells were focally embedded in the desmoplastic stroma. Focal stromal myxoid changes were observed in rare samples (7.7%, 4/52), in which the tumor cells showed cords or reticular patterns. Two samples (3.8%) exhibited pale cytoplasm. Obvious abundant neutrophil infiltration was observed in 19.2% (10/52) of the samples. Epithelioid ovoid or polygonal cells with moderate to high amounts of eosinophilic to clear cytoplasm were observed in most of the samples, and the nuclei were round, mildly pleomorphic, usually large and irregular with vesicular chromatin and prominent nucleoli. The tumor cells were cohesive (Fig. 2B).

Of the 20 thoracic SMARCA4-UTs, 100% presented a pure solid architecture. Necrosis was ubiquitously observed in eight (40%) samples. Eleven (55%) samples exhibited complete or focal rhabdoid morphology (Fig. 2G). The tumor cells had an epithelioid morphology and were poorly cohesive or noncohesive (Fig. 2F). Focal stromal myxoid changes were observed in one (5%) sample. An obvious acute inflammatory host response (abundant neutrophil infiltration) was observed in three (15%) samples.

Immunohistochemical results

Staining demonstrated BRG1-deficient expression in tumor cells, while inflammatory cells, which were used as internal positive controls, were positive for BRG1 expression (Figs. 2D and 2H). Complete loss of BRG1 expression in 97.2% of the samples (70/72) and diffuse severe reduction in BRG1 expression in >90% of the cells in 2.8% (2/72) of samples. The immunohistochemical staining results are summarized in Table 3 and Fig. 3. SMARCA4-dNSCLC expressed at least one epithelial marker (pan-cytokeratin and/or EMA) with diffuse and/or strong positive expression. However, pan-cytokeratin and/or EMA was scattered weakly expressed or not expressed by thoracic SMARCA4-UT.

Table 3 Immunohistochemical staining between SMARCA4-dNSCLC and thoracic SMARCA4-UT.

Immunohistochemical staining (positive expression (n, (%))	SMARCA4-dNSCLC	Thoracic SMARCA4-UT	p value	
Diffuse and/or strong positive expression of PCK and/or EMA	52 (100)	0		
CD34	1/35 (2.9)	14/15 (93.3)	<0.001*	
Syn	7/38 (18.4)	10/16 (62.5)	0.001*	
SALL4	2/28 (7.1)	3/13 (23.1)	0.147	
Notes:

SMARCA4-dNSCLC, SMARCA4-deficient non-small cell lung cancer; SMARCA4-UT, SMARCA4-deficient undifferentiated tumor; PCK, Pan-cytokeratin; EMA, epithelial membrane antigen; Syn, Synaptophysin; SALL4, spalt-like transcription factor 4.

* Statistically significant.

Figure 3 Immunohistochemical profile of thoracic SMARCA4-deficient tumors.

SMARCA4-dNSCLCs exhibited diffuse and strong expression of epithelial markers, including pan-cytokeratin (PCK) and epithelial membrane antigen (EMA). CK7 expression was diffuse and strong in lung adenocarcinomas. Thoracic SMARCA4-UTs exhibited scattered weak or negative expression for PCK and EMA. Most thoracic SMARCA4-UTs showed positive expression for the stem cell marker CD34 and the neuroendocrine marker synaptophysin (Syn), and some thoracic SMARCA4-UTs showed positive expression for the stem cell marker spalt-like transcription factor 4 (SALL4) . SMARCB1 (INI1) was retained expression in thoracic SMARCA-DTs. (magnification 200×). Thoracic SMARCA4-UT exhibits a distinctive uniform immunophenotype, with a panel of scattered weak or negative epithelial marker expression, deficient expression for BRG1, and positive expression for CD34 and Syn. This immunohistochemistry panel is helpful for quickly and accurately distinguishing thoracic SMARCA4-UT from SMARCA4-dNSCLC.

Expression of the stem cell marker CD34 and the neuroendocrine marker Syn was specific to SMARCA4-dNSCLC and thoracic SMARCA4-UT. Positive CD34 expression was observed in 2.9% (1/35) of the SMARCA4-dNSCLCs and 93.3% (14/15) of the thoracic SMARCA4-UTs (p < 0.001). Positive Syn expression was observed in 18.4% (7/38) of the SMARCA4-dNSCLCs and 62.5% (10/16) of the thoracic SMARCA4-UTs (p = 0.001). The stem cell marker spalt-like transcription factor 4 (SALL4) was expressed in a subset of samples, including SMARCA4-dNSCLCs (7.1%, 2/28) and thoracic SMARCA4-UTs (23.1%, 3/13) (p = 0.147). Therefore, thoracic SMARCA4-UT exhibits a distinctive uniform immunophenotype that was characterized by a combination of the scattered weak expression or negative expression of epithelial markers, deficient expression of BRG1, and positive expression of CD34 and Syn. This immunohistochemistry panel is helpful for quickly and accurately distinguishing thoracic SMARCA4-UT from SMARCA4-dNSCLC. The expression of the Ki67 proliferation index ranged from 30% to 90%. SMARCB1 (INI-1) expression was present in the available samples (43/43). All the available tested thoracic SMARCA4-UT samples were negative for lung adenocarcinoma markers (CK7, TTF-1, and Napsin A) and for squamous cell carcinoma markers (p63, p40, and CK5&6). In SMARCA4-dNSCLC samples, TTF-1, Napsin A, p63, p40, and CK5/6 were expressed in 5/50, 4/38, 6/42, 2/35 and 4/49 samples, respectively. All the tested samples were negative for nuclear protein in the testis (NUT) expression, which is used to diagnose NUT-middle carcinoma, and for targetable oncogenic therapy protein (ALK-V and ROS-1) expression.

Discussion

Thoracic SMARCA4-UT has been defined as a new entity of a rare high-grade malignant neoplasm that displays a deficiency of SMARCA4 (BRG1) (Nicholson et al., 2021; Decroix et al., 2020). However, SMARCA4 mutations can occur in approximately 5–16% of primary lung cancers (Orvis et al., 2014; Reisman et al., 2003) and in approximately 10–15% of lung adenocarcinomas (Reisman et al., 2003; Hoffman et al., 2014). Perret et al. (2019) described 30 patients with SMARCA4-deficient thoracic sarcoma ranging from 28 to 90 years of age (median: 48 years); this patient population had a marked male predominance (male: female = 9:1), and these patients were mostly smokers. Thoracic SMARCA4-DTs predominantly occurred in male patients who smoked. SMARCA4-dNSCLC and thoracic SMARCA4-UT have overlapping clinical characteristics, including history of smoking, male predominance and aggressive behavior. In our cohort, all the patients with thoracic SMARCA4-UT were male and had a history of smoking. A total of 98.1% of the patients with SMARCA4-dNSCLC were male, and 85.7% were either current or former smokers. The median ages at diagnosis were 61.0 years for patients with thoracic SMARCA4-UT and 59.5 years for patients with SMARCA4-dNSCLC. A high percentage of poorly differentiated NSCLC was also deficient in BRG1 expression (Yoshimoto et al., 2015). The percentage of patients with BRG1 deficiency was significantly greater in the large cell carcinoma and pleomorphic carcinoma group (13/38, 34.2%) than in the squamous cell carcinoma-adenocarcinoma group (7/95, 7.4%), indicating that a greater percentage of patients with poorly differentiated NSCLC had tumors with BRG1 deficient expression (Yoshimoto et al., 2015). In our cohort, all SMARCA4-dNSCLCs were poorly differentiated, and adenocarcinoma (59.6%) was the predominant histopathological type, followed by large cell carcinoma (15.4%).

SMARCA4-dNSCLC and thoracic SMARCA4-UT have overlapping pathological features. Morphologically, the typical features include a solid architecture with large, epithelioid cells, prominent nucleoli, and necrosis (Crombe et al., 2019; Perret et al., 2019; Sauter et al., 2017; Yoshida et al., 2017). In the present study, a pure solid architecture was the predominant feature of thoracic SMARCA4-DTs. Necrosis was ubiquitously observed in 46.2% of the SMARCA4-dNSCLCs and 40% of the thoracic SMARCA4-UTs. Epithelial architecture (focal glandular formation) was observed in only 7.7% of the SMARCA4-dNSCLC samples. Rhabdoid tumor cells are an important morphological feature. 55% of the thoracic SMARCA4-UTs exhibited a rhabdoid morphology, which was more than the 19.2% of the SMARCA4-dNSCLCs that exhibited this morphology. Epithelial architecture and cellular cohesion are helpful for diagnosing these two forms of cancer (Nicholson et al., 2021). Tumor cells were cohesive in the SMARCA4-dNSCLCs, whereas, tumor cells were poorly cohesive or noncohesive in the thoracic SMARCA4-UTs. SMARCA4-dNSCLC exhibits the differentiation of epithelial architecture, and the focal glandular formation can be observed in a few SMARCA4-deficient lung adenocarcinomas. Histological distinction between these two entities is difficult, and their differential diagnosis is challenging, especially on only small biopsies. Therefore, in routine pathological diagnosis, immunohistochemical staining for BRG1 and epithelial makers should be performed to diagnose thoracic cancers with solid architecture and/or necrosis and with poorly differentiated and undifferentiated morphology, especially those with rhabdoid morphology.

SMARCA4-dNSCLC has unique clinicopathological features and a worse prognosis than SMARCA4-intact NSCLC (Bell et al., 2016; Liang et al., 2023). Thoracic SMARCA4-UT tends to have larger primary tumor size and to be associated with worse survival than SMARCA4-dNSCLC (Rekhtman et al., 2020). SMARCA4-dNSCLC can be distinguished from thoracic SMARCA4-UT according to their gland architecture, cellular cohesion and expression of epithelial markers. SMARCA4-dNSCLC are characterized by the diffuse and/or strong expression of epithelial markers. However, epithelial markers exhibited only scattered weak expression or negative expression in thoracic SMARCA4-UT. The immunophenotype of thoracic SMARCA4-UT is quite specific, and most thoracic SMARCA4-UTs express CD34 and Syn. Some thoracic SMARCA4-UTs have a neuroendocrine-like phenotype and may mimic LCNEC with rhabdoid features. LCNEC with rhabdoid features often diffusely expresses the neuroendocrine markers Syn, CgA and CD56, and tumor cells express epithelial markers. Thoracic SMARCA4-UT may show variable degrees expression of Syn, but none of the samples expressed more than one neuroendocrine marker. The expression of INI-1 (SMARCB1), another core regulatory subunit in the SWI/SNF complex, was preserved in all the available samples.

Thoracic SMARCA4-DTs have an aggressive clinical course and a poor prognosis. BRG1 plays roles in carcinogenesis and tumor progression (Masliah-Planchon et al., 2015; Halliday et al., 2009), and BRG1 deficiency is associated with highly aggressive behavior and poor survival (Bell et al., 2016; Liang et al., 2023; Reisman et al., 2003). In the present series, most patients with SMARCA4-dNSCLC and thoracic SMARCA4-UT were in an advanced stage of disease and had a worse survival, with a median OS of 7.8 months for SMACAR4-dNSCLCs and 5.6 months for thoracic SMARCA4-UTs. Among the tumors, 93.75% of the thoracic SMARCA4-UTs and 79.5% of the SMARCA4-dNSCLCs were staged as TNM III–IV. Due to the short-term follow-up and limited number of patients, OS and PFS were not significantly different between patients with SMARCA4-NSCLC and patients with thoracic SMARCA4-UT in the present study. Long-term follow-up of more thoracic SMARCA4-DT patients is needed. SMARCA4-dNSCLC and thoracic SMARCA4-UT have overlapping clinical characteristics, pathological features and poor prognosis. We hypothesize that thoracic SMARCA4-UT may represent the undifferentiated or dedifferentiated form of SMARCA4-dNSCLC. However, further studies on the differentiation of thoracic SMARCA4-DTs in larger series are needed for validation.

No specific therapeutic strategies have been developed for SMARCA4-deficient neoplasms. Chemotherapy is a common treatment for thoracic SMARCA4-DTs, but the response of these patients to conventional chemotherapy is limited. Immunotherapy has been shown to be effective in few reported patients (Anzic et al., 2021; Tanaka et al., 2021; Shinno et al., 2022); however, limited efficacy has been shown in some patients (Gantzer et al., 2022). SMARCA4 deficiency was shown to be synthetically lethal with CDK4/6 inhibition in NSCLC, and CDK4/6 inhibitors may be effective in this significant subgroup of NSCLC patients (Xue et al., 2019). SMARCA4-inactivating mutations increase sensitivity to the Aurora kinase A inhibitor VX-680 in NSCLC (Tagal et al., 2017). A potent and selective EZH2 inhibitor currently in phase II clinical trials induces potent antiproliferative and antitumor effects in SCCOHT cell lines and xenografts deficient in both SMARCA2 and SMARCA4 (Chan-Penebre et al., 2017). Potent inhibitors of enhancer of zeste homolog 2 (EZH2) or histone deactylase may seem promising and effective (Chan-Penebre et al., 2017; Yamagishi & Uchimaru, 2017; Leitner et al., 2020). However, further studies on immunotherapy and potent therapies for the treatment of thoracic SMARCA4-DTs are needed.

Conclusions

Thoracic SMARCA4-DTs include SMARCA4-dNSCLC and thoracic SMARCA4-UT, which exhibit solid architecture, necrosis and/or rhabdoid morphology. Both have overlapping clinicopathological features and a poor prognosis. Thoracic SMARCA4-UT is characterized by an immunohistochemical panel of scattered weak or negative expression of epithelial markers and positive expression of CD34 and Syn. We hypothesize that thoracic SMARCA4-UT may be the undifferentiated or dedifferentiated form of SMARCA4-dNSCLC, but additional exploration is needed for validation in a larger series.

Supplemental Information

Supplemental Information 1 Raw data of SAMRCA4 deficient thoracic tumor.

Click here for additional data file.

Abbreviations

SMARCA4-DT SMARCA4-deficient tumor

SMARCA4-dNSCLC SMARCA4-deficient non-small cell lung cancer

SMARCA4-UT SMARCA4-deficient undifferentiated tumor

SWI/SNF SWItch/Sucrose Non-Fermentable

BRG1 Brahma-related gene 1

PCK Pan-cytokeratin

EMA epithelial membrane antigen

Syn Synaptophysin

SALL4 spalt-like transcription factor 4

Additional Information and Declarations

Competing Interests

Author Contributions

Human Ethics

Data Availability

The authors declare that they have no competing interests.

Ping Zhou conceived and designed the experiments, performed the experiments, analyzed the data, prepared figures and/or tables, authored or reviewed drafts of the article, and approved the final draft.

Yiyun Fu performed the experiments, analyzed the data, prepared figures and/or tables, and approved the final draft.

Yuan Tang analyzed the data, prepared figures and/or tables, and approved the final draft.

Lili Jiang conceived and designed the experiments, authored or reviewed drafts of the article, and approved the final draft.

Weiya Wang conceived and designed the experiments, authored or reviewed drafts of the article, and approved the final draft.

The following information was supplied relating to ethical approvals (i.e., approving body and any reference numbers):

Ethics approval was obtained from the respective ethics committees of West China Hospital, Sichuan University, China (NO.2022317).

The following information was supplied regarding data availability:

The raw data of SMARCA4 deficient thoracic tumors is available in the Supplemental File.

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
