# Peer review of "Thoracic SMARCA4-deficient tumors: a clinicopathological analysis of 52 cases with SMARCA4-deficient non-small cell lung cancer and 20 cases with thoracic SMARCA4-deficient undifferentiated tumor"

_PeerJ, doi:10.7717/peerj.16923_

## Round 0.1 · original submission · Major Revisions

Please answer the reviewer's questions carefully and try your best to revise the manuscript.

**Language Note:** The review process has identified that the English language must be improved. PeerJ can provide language editing services - please contact us at copyediting@peerj.com for pricing (be sure to provide your manuscript number and title). Alternatively, you should make your own arrangements to improve the language quality and provide details in your response letter. – PeerJ Staff

Reviewer 1 ·

Basic reporting

This paper compares the clinicopathological, immunohistochemical, and molecular entities between SMARCA4-deficient NSCLC (SMARCA4-dNSCLC) and SMARCA4-UT. The authors were able to determine that SMARCA4-dNSCLC can be distinguished from thoracic SMARCA4-UT by gland architecture, diffuse strong keratin expression, and immunohistochemistry. The work appears to be well done.

Experimental design

I do not have any major criticisms for this paper.

Minor comments:

The structure of sentences can be improved. In addition, the legends are at times incomplete which makes it difficult to evaluate the paper. For example, they should describe the scale bars, and the markers used for staining. In the methods, it is unclear which protocol has been used for the immunohistochemistry.

Validity of the findings

To investigate the difference in the characteristics of SMARCA4-deficient tumors is important to better characterize the role of this gene in tumorigenesis.

Reviewer 2 ·

Basic reporting

The manuscript requires substantial grammatical revisions throughout the entire text. While the research idea presents an innovative approach to promoting a new IHC staining panel for distinguishing subtypes in lung carcinoma, the authors need to strengthen their arguments supporting the rationale behind this research. Moreover, they should emphasize the innovation and significance of their findings, as the core importance of these findings has not been adequately addressed.

1. Grammar:
Major grammatical revisions are needed throughout the manuscript. Refer to remarks in the annotated text.
Rewriting is necessary for the introduction lines 53-77 to simplify the text and enhance coherence for the reader. Review comments in the annotated text.
Consider using an AI-based language model to refine the text flow and ensure appropriate English usage.
2. Introduction:
Enhance the introduction by providing descriptions of the markers used in IHC panels, including the full name and abbreviation for each marker. Additionally, justify the importance of each marker in cancer/pathological screens with references to relevant literature.
3. Clearly state the hypothesis and elaborate on what the authors hope to find, emphasizing the importance of the research (line 77-78).

Experimental design

Within the Aims and Scope of the journal, the primary research question needs better definition.
4. Expand the methods section, particularly regarding technical information related to IHC. See comments in the annotated manuscript text.

Refine the presentation of data, addressing grammatical issues, figure structure, and specific subfigure references (as noted in the annotated manuscript).
5. Figure 2:
To the figure images add the protein name used for staining. (ie; BRG1...)
In the figure legend, include the full name of the antibodies used in the staining.
Enhance Figure 2 by adding enlarged images (cropped and magnified) of positive staining, with arrows indicating positive or diffused staining.
6. Figure 2 Results Text:
Reference Figure 2 (C/D/G/H) in the results text to direct readers to the relevant panel. (Figure 2 panel C/D/G/H were not referenced in the text at all)
7. Results (Line 173): When making comparisons between dsNSCLC and UT there is no data presented to back up the claim. Add a dsNSCLC panel stained with CD34/SYN/SALL4 to Figure 3 to demonstrate comparative results.

Validity of the findings

Strengthen the link between the conclusions and the original research question.
Emphasize the importance and innovation of the findings, providing a clearer connection to the research's significance.

Annotated reviews are not available for download in order to protect the identity of reviewers who chose to remain anonymous.

Reviewer 3 ·

Basic reporting

This study shows thoracic SMARCA4-deficient tumors, including SMARCA4-deficient NSCLC and SMARCA4-deficient undifferentiated tumors, present significant clinicopathologic overlap as aggressive poorly differentiated malignancies predominantly in male smokers. However, NSCLC maintained glandular architecture and diffuse keratin expression distinguishing them from entirely solid and keratin-negative undifferentiated counterparts. Despite no differences in survival, classification via histological and immunohistochemical features remains important given emerging differences in molecular profiling and targeted therapeutic approaches for these SMARCA4-deficient entities.

Overall the study design and findings appear novel and meaningful. The authors should address the following points to further strengthen the quality before the paper is considered for publication.

Major points:
1. Provide more details on the patient cohort - how were cases diagnosed and selected? Whether patients with tumors located in the lung or thorax were diagnosed with carcinoma in situ? Additionally, articulate the exclusion criteria, including cases where BRG1 expression is normal or lacks immunostaining results for BRG1? Were certain histological types excluded? This will clarify any sampling bias.

2. Expand the methods section with more specifics on the statistical tests used for survival analysis, immunohistochemistry quantification, and significance values.

3. Incorporate some explanatory discussion of the overlapping features and differences between SMARCA4-dNSCLC and SMARCA4-UT. Are there hypotheses for the underlying reasons behind the distinguishing characteristics?

Minor points:
1. In line 67, the number is wrong regarding the 11% (188 of 27,281).
2. In line118 and line 128, In Table, the values 59.9 and 61.0 correspond to the median age. However, the result text inaccurately describes them as the average age.
3. In the table, the total number of patients varies across different analyses. This should be explicitly addressed and explained in the article.
4. Carefully proofread the manuscript to resolve any outstanding grammar issues and typographical errors.

Experimental design

As I mentioned before, the author should provide more details on the patient cohort.

Validity of the findings

No comment

---

## Round 0.2 · accepted · Accept

The manuscript was revised by the authors, and two reviewers agreed to accept it. I also reviewed the manuscript and found no obvious publication risk; therefore, I approved the manuscript for publication.

Reviewer 2 ·

Basic reporting

The proposed corrections have been skillfully integrated by the authors, resulting in manuscripts and data that are now more transparent and user-friendly. The implemented revisions have not only clarified the content but also made it more readily understandable for the readers. This thoughtful incorporation of suggested changes enhances the overall quality of the material.

Experimental design

.

Validity of the findings

.

Reviewer 3 ·

Basic reporting

The authors thoroughly addressed all of my inquiries and meet the criteria for publication.

Experimental design

no comment

Validity of the findings

no comment